# A Local Method for Satisfying Interventional Fairness with Partially Known Causal Graphs

**Haoxuan Li**[1,2]    **Yue Liu**[3,*]    **Zhi Geng**[4]    **Kun Zhang**[2,5]
[1]Peking University    [2]Mohamed bin Zayed University of Artificial Intelligence
[3]Renmin University of China    [4]Beijing Technology and Business University
[5]Carnegie Mellon University
hxli@stu.pku.edu.cn, liuyue_stats@ruc.edu.cn
zhigeng@btbu.edu.cn, kunz1@cmu.edu

## Abstract

Developing fair automated machine learning algorithms is critical in making safe and trustworthy decisions. Many causality-based fairness notions have been proposed to address the above issues by quantifying the causal connections between sensitive attributes and decisions, and when the true causal graph is fully known, certain algorithms that achieve interventional fairness have been proposed. However, when the true causal graph is unknown, it is still challenging to effectively and efficiently exploit partially directed acyclic graphs (PDAGs) to achieve interventional fairness. To exploit the PDAGs for achieving interventional fairness, previous methods have been built on variable selection or causal effect identification, but limited to reduced prediction accuracy or strong assumptions. In this paper, we propose a general min-max optimization framework that can achieve interventional fairness with promising prediction accuracy and can be extended to maximally oriented PDAGs (MPDAGs) with added background knowledge. Specifically, we first estimate all possible treatment effects of sensitive attributes on a given prediction model from all possible adjustment sets of sensitive attributes via an efficient local approach. Next, we propose to alternatively update the prediction model and possible estimated causal effects, where the prediction model is trained via a min-max loss to control the worst-case fairness violations. Extensive experiments on synthetic and real-world datasets verify the superiority of our methods. To benefit the research community, we have reAleased our project at https://github.com/haoxuanli-pku/NeurIPS24-Interventional-Fairness-with-PDAGs.

## 1 Introduction

Making automated machine learning algorithms fair is critical to producing safe and trustworthy decisions with different sensitive attributes [Brennan et al., 2009, Dieterich et al., 2016, Agarwal et al., 2018, Chen et al., 2018, Chouldechova et al., 2018, Hoffman et al., 2018, Yurochkin et al., 2019, Li et al., 2023]. To achieve fair predictions, association-based and causality-based fairness notions have been proposed. Specifically, the former requires statistical independence between the sensitive attribute and predicted outcome [Dwork et al., 2012, Hardt et al., 2016, Chouldechova, 2017, Jin et al., 2024a,b], whereas the later investigates causal effect of the sensitive attribute on the predicted outcome, requiring that the predicted outcome be the same across the real-world without intervention and the counterfactual world with intervention on sensitive attribute [Zhang and Bareinboim, 2018, Zhang et al., 2017a,b, 2018a,b, Khademi et al., 2019, Galhotra et al., 2022].

---

[*]Yue Liu is the corresponding author.

Despite many algorithms have been developed to achieve causality-based fairness, most of them require the true causal directed acyclic graph (DAG) is fully known [Kusner et al., 2017, Nabi and Shpitser, 2018, Chiappa, 2019, Chikahara et al., 2021]. Nevertheless, true causal DAGs and structural equations are usually not directly available in practice [Colombo et al., 2014]. Moreover, without strong assumptions, e.g., linearity [Shimizu et al., 2006] and additive noise [Hoyer et al., 2008, Peters et al., 2014], the true causal DAG may not be recoverable from only the observed data, which raises a great challenge to achieve causality-based fairness based on partially DAGs (PDAGs).

To achieve causality-based fairness under partially known causal graphs, recent approaches can be broadly classified into two categories: variable selection methods [Zuo et al., 2022] and causal effect identification methods [Perkovic, 2020, Zuo et al., 2024]. Specifically, the variable selection method first adopts causal discovery algorithms to obtain a Markov equivalence class of DAGs that encode the same set of conditional independencies from the data, and then classifies the covariate variables into three categories: definite non-descendants, possible descendants, and definite descendants of the sensitive attributes. By noting that a prediction model would be counterfactually fair if the prediction model is a function of the non-descendants of sensitive attributes [Kusner et al., 2017], these methods proposed to use only the definite non-descendants or further incorporate possible descendants to make fair predictions. Despite can theoretically guarantee interventional fairness, disregarding the descendants results in a notable decline in performance. Another category of methods proposed to identify causal effects of sensitive attributes on the outcome variable directly from the maximally oriented PDAGs (MPDAGs), but relies on strong assumptions for identification.

In this paper, we aim to effectively and efficiently achieve interventional fairness with partially known causal graphs. Different from the previous variable selection and causal effect identification methods, we exploit all variables to ensure relatively high prediction accuracy, as well as does not need to rely on additional strong assumptions for identification. Specifically, we propose a novel local method to partially identify the causal effects of sensitive attributes on the predictor for satisfying the interventional fairness, which does not require a global search of all possible DAGs, but can estimate all possible causal effects using the obtained CPDAG. Inspired by the IDA framework [Maathuis et al., 2009], we first propose a local algorithm to obtain possible parental sets of the sensitive attributes on the PDAGs, from which we estimate all possible propensities for various cases. We then calculate all possible violations of intervention fairness using all possible propensities. Next, we propose to alternatively update the prediction model and the corresponding estimation of the possible causal effects, where the prediction model is trained via a min-max loss to control the worst-case fairness violations. The validity of our method also holds for MPDAGs with added background knowledge.

The contributions of this paper are summarized as follows:

• We propose a general min-max optimization framework to achieve interventional fairness, which enables to use all variables to achieve relatively high prediction accuracy, and can be extended to MPDAGs with added background knowledge.

• Based on the proposed framework, we provide an efficient algorithm to estimate all possible causal effects of sensitive attribute on predictions for MPDAGs.

• We further provide a joint learning approach that alternatively updates the prediction model and the corresponding estimation of the possible causal effects, where the prediction model is trained via a min-max loss to control the worst-case fairness violations.

• We conduct extensive experiments on synthetic and real-world datasets to demonstrate the effectiveness of our methods in achieving interventional fairness with promising accuracy.

## 2 Preliminaries

### 2.1 DAGs, PDAGs, CPDAGs, and MPDAGs

In a graph $\mathcal{G} = (V, E)$, where $V$ and $E$ represent the node set and edge set in $\mathcal{G}$, we say $\mathcal{G}$ is *directed*, *undirected*, or *partially directed* if all edges in the graph are directed, undirected, or a mixture of directed and undirected edges, respectively. The *skeleton* of $\mathcal{G}$ is an undirected graph obtained by removing all arrowheads from $\mathcal{G}$. Given a graph $\mathcal{G}$, an $X_i$ is called a *parent* of $X_j$ and $X_j$ is called a *child* of $X_i$ if $X_i \rightarrow X_j$ in $\mathcal{G}$. Also, $X_i$ is a *sibling* of $X_j$ if $X_i - X_j$ in $\mathcal{G}$. If $X_i$ and $X_j$ are connected by an edge, they are *adjacent*. The notation $pa(X_i, \mathcal{G})$, $ch(X_i, \mathcal{G})$, $sib(X_i, \mathcal{G})$, and $adj(X_i, \mathcal{G})$ respectively represent sets of parents, children, siblings, and adjacent vertices of $X_i$ in

$\mathcal{G}$. A graph is termed *complete* if all distinct vertices are adjacent. A *path* is a sequence of distinct vertices $(X_{k_1}, \cdots, X_{k_j})$ where any two consecutive vertices are adjacent. A path is called *partially directed* from $X_{k_1}$ to $X_{k_j}$ if $X_{k_i} \leftarrow X_{k_{i+1}}$ does not occur in $\mathcal{G}$ for any $i = 1, \ldots, j-1$. A partially directed path is *directed* (*undirected*) if all edges on the path are directed (undirected). A vertex $X_i$ is an *ancestor* of $X_j$ and $X_j$ is a *descendant* of $X_i$ if there is a directed path from $X_i$ to $X_j$ or $X_i = X_j$. A directed (undirected) *cycle* is a directed (undirected) path from a vertex to itself. Particularly, a cycle with the number of edges equal to three is called a *triangle*.

In a directed acyclic graph (DAG), all edges are directed and there is no directed cycle. A partially directed acyclic graph (PDAG) may contain both directed and undirected edges without directed cycles. Two DAGs are Markov *equivalent* if they induce the same set of conditional independence relations [Pearl, 1988]. A *Markov equivalence class*, denoted by $[\mathcal{G}]$, contains all DAGs equivalent to $\mathcal{G}$. A Markov equivalence class can be uniquely represented by a partially directed graph called *completely partially directed acyclic graph* (CPDAG) $\mathcal{G}^*$, in which two vertices are adjacent if and only if they are adjacent in $\mathcal{G}$, and a directed edge occurs if and only if it appears in all DAGs in $[\mathcal{G}]$ [Andersson et al., 1997, Chickering, 2002a]. A CPDAG $\mathcal{G}^*$ can be refined to a maximally oriented partially directed acyclic graph (maximal PDAG or MPDAG) $\mathcal{H}$ by giving background knowledge $\mathcal{B}_d$ consisting of some directed causal relationships between variables in the form $X_i \rightarrow X_j$ [Hauser and Bühlmann, 2012, Eigenmann et al., 2017, Wang et al., 2017, Rothenhäusler et al., 2018]. Meek [1995] proved that, with a series of orientation rules called Meek's rules, some undirected edges may be further directed (see Algorithm 4 in Appendix for details), and the resulting graph is an MPDAG. Both a DAG and a CPDAG can be viewed as special cases of an MPDAG, where the background knowledge is fully known and unknown, respectively.

## 2.2 Structural Causal Model

We follow Pearl [2009] to define the structural causal model (SCM) as a triplet $(V, U, F)$ to describe the causal relationships between variables. Specifically, $V$ is a set of observable endogenous variables, $U$ is a set of latent independent background variables that cannot be caused by any variable in $V$, and $F$ is a set of functions $\{f_1, \ldots, f_{|V|}\}$, one for each $V_i \in V$, such that $V_i = f_i(pa_i, U_i)$, where $pa_i \subseteq V \setminus \{V_i\}$ and $U_i \in U$. Notably, the set of equations $F$ induces a directed graph over the variables, here assumed to be a DAG, where the directed causes of $V_i$ represents its parent set. A causal DAG model consists of a DAG $\mathcal{G}$ and a joint distribution $P$ over $V$ such that the distribution can be factorized as $P(v_1, ..., v_{|V|}) = \prod_{v_i \in V} P(v_i | pa(v_i, \mathcal{G}))$ [Perković et al., 2017].

## 2.3 Interventional Fairness

Given a DAG $\mathcal{G}$ and two distinct variables $X$ and $Y$, the causal effect of $X$ on $Y$ can be interpreted by the post-intervention distribution of $Y$ intervening on $X$ via *do* operator [Pearl, 1995, 2009]. Formally, given a distribution $P(U)$ over the background variables $U$, an intervention $do(X = x)$ that force variable $X$ to take certain value $x$ is defined as the substitution of the structural equation $X = f_x(pa_x, U_x)$ with $X = x$, and the post-interventional density of $Y$ is denoted as $f(Y = y \mid do(X = x))$. However, if we only know a CPDAG $\mathcal{G}^*$ or an MPDAG $\mathcal{H}$, the causal effect of $X$ on $Y$ may not be identifiable from observational data [Perković et al., 2015, 2017, Wu et al., 2019a,b].

Build on the *do* operator, interventional fairness criterion [Kilbertus et al., 2017, Salimi et al., 2019] requires that given the covariates, intervening the value of the sensitive attribute does not affect the output distribution of the output predictor. Formally, let $A$, $Y$, and $X$ denote sensitive attributes, outcomes of interest, and other covariates, and $\hat{Y}$ be a predictor produced by a learning algorithm as a prediction of $Y$. Without loss of generality[2], we say the predictor $\hat{Y}$ is interventionally fair with respect to the sensitive attributes $A$ if it satisfies the following condition:

**Definition 2.1** (Interventional fairness [Kilbertus et al., 2017]). We say the prediction $\hat{Y}$ is interventionally fair with respect to the sensitive attributes $A$ if the following holds:

$$P(\hat{Y} = y \mid do(A = a)) = P(\hat{Y} = y \mid do(A = a')),$$

for all possible values of $y$ and any value that $A$ can take.

---

[2]Note that our proposed method can be naturally extend to interventional fairness with admissible attributes $X_{ad} \subseteq X$ [Salimi et al., 2019], defined as $P(\hat{Y} = y \mid do(A = a), do(X_{ad} = x_{ad})) = P(\hat{Y} = y \mid do(A = a'), do(X_{ad} = x_{ad}))$, by observing that $do(A)$ and $do(X_{ad})$ are symmetric and including $X_{ad}$ into $A$.

# 3 A General Min-Max Optimization Framework

## 3.1 Motivation and Method Overview

Given only observational data, the underlying true causal DAG may not be recoverable without strong assumptions such as linearity [Shimizu et al., 2006] or additive noise [Hoyer et al., 2008, Peters et al., 2014]. Instead, we can use causal discovery algorithms [Spirtes and Glymour, 1991, Shimizu et al., 2006, Zhang and Hyvärinen, 2009, Peters et al., 2014] to obtain a CPDAG that contains that true causal DAG. To exploit the obtained CPDAG for achieving interventional fairness, previous methods have been built on variable selection [Zuo et al., 2022] or causal effect identification [Perkovic, 2020, Zuo et al., 2024], but limited to reduced prediction accuracy or strong assumption for identification.

Differing from the above work, we propose a novel local method to partially identify the causal effects of sensitive attributes on the predictor for satisfying the interventional fairness. Interestingly, our approach does not require a global search of all possible DAGs, but can estimate all possible causal effects using the obtained CPDAG. In the following, we first theoretically state the necessary and sufficient condition for discriminating a set to be a possible parent set of the sensitive attribute for CPDAG and MPDAG, respectively, from which we propose a local method for finding all possible parent sets of sensitive attributes and estimating the corresponding propensities (Sec. 3.2). We then calculate the all possible degrees of intervention fairness being violated using all possible estimated propensities (Sec. 3.3). Finally, we further propose a min-max joint learning approach to make the predictor satisfy intervention fairness by controlling for worst-case fairness violation (Sec. 3.4).

## 3.2 Finding Possible Parental Sets and Estimating Propensities

Given a CPDAG obtained from observational data, since enumerating all DAGs is infeasible when the size of the Markov equivalence class is large [He et al., 2015, Zuo et al., 2022], we propose to adopt a novel framework called IDA [Maathuis et al., 2009, Fang and He, 2020] to only enumerate possible parental sets of the sensitive attribute. This provides a more efficient solution since enumerating possible parental sets only requires the local structure around the sensitive attribute. We further extend the above theoretical results to MPDAGs with background knowledge added, and estimate possible propensities by regressing sensitive attribute on each possible parental set.

**Definition 3.1** (v-structure). For three distinct vertices $X_i, X_j$ and $X_k$, if $X_i \to X_j \leftarrow X_k$ and $X_i$ is not adjacent to $X_k$ in $\mathcal{G}$, then the triplet $(X_i, X_j, X_k)$ is called a v-structure collided on $X_j$.

Pearl [2009] have shown that two DAGs are equivalent if and only if they have the same skeleton and the same *v-structures*. Given a CPDAG $\mathcal{G}^*$ contains all DAGs equivalent to $\mathcal{G}$ and a sensitive attribute $A$, the local structure around $A$ can be divided into three cases: parents $pa(A, \mathcal{G}^*) \to A$, children $pa(A, \mathcal{G}^*) \leftarrow A$, and siblings $sib(A, \mathcal{G}^*) - A$ with undirected edges. Let $\mathbf{S}(A)$ be a subset of $sib(A, \mathcal{G}^*)$, we denote $\mathcal{G}^*_{\mathbf{S}(A) \to A}$ as a DAG that is obtained from CPDAG $\mathcal{G}^*$ by changing all undirected edges $\{Z - A, \forall Z \in \mathbf{S}(A)\}$ into the directed edges $\{Z \to A, \forall Z \in \mathbf{S}(A)\}$ as parents, and all of other undirected edges $\{Z - A, \forall Z \notin \mathbf{S}(A)\}$ into the directed edges with opposite direction $\{Z \leftarrow A, \forall Z \notin \mathbf{S}(A)\}$ as children. We say $\mathbf{S}(A) \to A$ is a possible parental set of the sensitive attribute $A$ for $\mathcal{G}^*$, if there exists a DAG $\mathcal{G}$ in the equivalence class $\mathcal{G}^*$ with the same directed edges adjacent to $A$ as $\mathcal{G}^*_{\mathbf{S}(A) \to A}$. Then a sufficient and necessary condition for determining whether a set $\mathbf{S}(A) \subset \text{sib}(A, \mathcal{G}^*)$ is a possible parent set of the sensitive attribute $A$ is shown in below.

**Lemma 3.2** (Maathuis et al. [2009]). *Given a CPDAG $\mathcal{G}^*$, a set $\mathbf{S}(A) \subset \text{sib}(A, \mathcal{G}^*)$ is a possible parent set of the sensitive attribute $A$, if and only if there is no more v-structure in $\mathcal{G}^*_{\mathbf{S}(A) \to A}$ than $\mathcal{G}^*$.*

From the above lemma, given any $\mathbf{S}(A) \subseteq sib(A, \mathcal{G}^*)$, we can determine whether $\mathbf{S}(A)$ is a possible parental set from a local way. In particular, let the *induced subgraph* of $\mathcal{G} = (V, E)$ over $V' \subseteq V$ be the subgraph $\mathcal{G}' = (V', E')$ by restricting the edges $E$ on the set of vertices $V'$, where the edge set $E'$ contains all edges with both endpoints in $V'$. Then Lemma 3.2 is equivalent to check whether the induced subgraph of $\mathcal{G}^*$ over $\mathbf{S}(A)$ is complete, i.e., all vertices in the induced subgraph of $\mathcal{G}^*$ over $\mathbf{S}(A)$ are adjacent. This is because if there are two vertices $X_i$ and $X_j$ in $\mathbf{S}(A)$ that are not adjacent, then by Definition 3.1, a v-structure is formed as $X_i \to A \leftarrow X_j$.

For MPDAG, a key difference compared with CPDAG is the possible generation of a directed triangular cycle (e.g., $A \to X_i \to X_j \to A$) [Fang and He, 2020], when incorporating the background

**Algorithm 1:** A local algorithm for finding possible adjustment sets and estimating corresponding propensity model parameters of the sensitive attribute $A$ further using direct causal information.

**Input:** Sensitive attribute $A$, CPDAG $\mathcal{G}^*$, and consistent direct causal information set $\mathcal{B}_d$.

1 Construct the MPDAG $\mathcal{H}$ from $\mathcal{G}^*$ and $\mathcal{B}_d$ using Meek's rules (see Algotirhm 4 for details);
2 Set $\mathcal{S}_A = \emptyset$ and $m = 1$;
3 **for** *each* $\mathbf{S}^{(m)} \subset \mathrm{sib}(A, \mathcal{H})$ *such that orienting* $\mathbf{S}^{(m)} \to A$ *and* $A \to \mathrm{sib}(A, \mathcal{H}) \backslash \mathbf{S}^{(m)}$ *does not introduce any v-structure collided on $A$ or any directed triangle containing $A$* **do**
4     **for** *number of steps for training the possible propensity model on* $\mathbf{S}^{(m)}$ **do**
5         Sample a batch of units $\{(a_{m_k}, x_{m_k}|_{\mathbf{S}^{(m)}})\}_{k=1}^{K}$;
6         Update $\hat{\phi}^{(m)}$ by descending along the gradient $\nabla_{\hat{\phi}^{(m)}} \ell(\hat{\phi}^{(m)}; \mathbf{S}^{(m)})$;
7     **end**
8     $\mathcal{S}_A \leftarrow \mathcal{S}_A \cup (pa(A, \mathcal{H}) \cup \mathbf{S}^{(m)})$ and $m \leftarrow m + 1$;
9 **end**

**Output:** A set $\mathcal{S}_A$ of possible adjustment sets $\mathbf{S}^{(m)}$ and propensity model parameters $\hat{\phi}^{(m)}$.

knowledge and using Meek's rule for orienting undirected edges adjacent to the sensitive attribute $A$. Motivated by such difference, we entend the theoretical results on CPDAGs to MPDAGs for determining possible parental sets of $A$, which can also be implemented via a local way.

**Definition 3.3** (Direct triangle structure)**.** For three distinct vertices $X_i, X_j$ and $X_k$, if $X_i \to X_j \to X_k \to X_i$, then the triplet $(X_i, X_j, X_k)$ is called a direct triangle structure.

**Lemma 3.4** (Fang and He [2020])**.** *Given an MPDAG $\mathcal{H}$, a set $\mathbf{S}(A) \subset \mathrm{sib}(A, \mathcal{H})$ is a possible parent set of $A$, if and only if there is no more direct triangle structure and v-structure in $\mathcal{H}_{\mathbf{S}(A) \to A}$ than $\mathcal{H}$.*

From Lemma 3.4, we can conclude that for a given MPDAG $\mathcal{H}$, it is equivalent to checking whether the induced subgraph of $\mathcal{H}$ over $\mathbf{S}(A)$ is complete, as well as there does not exist $S \in \mathbf{S}(A)$ and $C \in adj(A, \mathcal{H}) \backslash (pa(A, \mathcal{H}) \cup \mathbf{S}(A))$ such that $C \to S$, otherwise a direct triangle structrue is formed as $A \to C \to S \to A$. This provides a efficient way to locally find the possible parental sets of $A$. Without loss of generality, we denote the set of possible parental sets of $A$ with a total number $M$ as

$$\mathcal{S}_A = \Big\{ pa(A, \mathcal{H}) \cup \mathbf{S}^{(1)}(A), pa(A, \mathcal{H}) \cup \mathbf{S}^{(2)}(A), \ldots, pa(A, \mathcal{H}) \cup \mathbf{S}^{(M)}(A) \Big\}.$$

Next, to estimate $P(\hat{Y} = y | do(A = a))$ in the interventional fairness notion, for each possible parental set $pa(A, \mathcal{H}) \cup \mathbf{S}^{(m)}(A)$ with $m = 1, \ldots, M$, we propose to first estimate the corresponding propensity $P(A \mid pa(A, \mathcal{H}) \cup \mathbf{S}^{(m)}(A))$. Specifically, we use the covariates $X$ restricted on $pa(A, \mathcal{H}) \cup \mathbf{S}^{(m)}(A)$, denoted as $X|_{pa(A, \mathcal{H}) \cup \mathbf{S}^{(m)}(A)}$, and train the propensity model $g(X|_{pa(A, \mathcal{H}) \cup \mathbf{S}^{(m)}(A)}; \hat{\phi}^{(m)})$ for estimating propensity $P(A \mid pa(A, \mathcal{H}) \cup \mathbf{S}^{(m)}(A))$ by minimizing

$$\ell(\hat{\phi}^{(m)}) = -\frac{1}{N} \sum_{i=1}^{N} \Big[ A_i \log g(X|_{pa(A, \mathcal{H}) \cup \mathbf{S}^{(m)}(A)}) + (1 - A_i) \log \big( 1 - g(X|_{pa(A, \mathcal{H}) \cup \mathbf{S}^{(m)}(A)}) \big) \Big],$$

where $\hat{\phi}^{(m)}$ is the propensity model parameter and $\hat{e}_i^{(m)} = g(x_i|_{pa(A, \mathcal{H}) \cup \mathbf{S}^{(m)}(A)}; \hat{\phi}^{(m)})$ is the estimated propensity of unit $i$ corresponding to the possible parent set $pa(A, \mathcal{H}) \cup \mathbf{S}^{(m)}(A)$ for $i = 1, \ldots, N$ and $m = 1, \ldots, M$. Since CPDAGs can be viewed as special cases of MPDAGs, without loss of generality, we summarized the proposed local algorithm on MPDAGs in Alg. 1 (see Alg. 3 in Appendix for implementing the proposed local algorithm on CPDAGs).

### 3.3 Estimating and Bounding Interventional Fairness

We then aim to estimate and bound all possible causal effects of sensitive attribute $A$ on the predictor $\hat{Y}$. Note that each parental set $pa(A, \mathcal{H}) \cup \mathbf{S}^{(m)}(A)$ is a valid back-door adjustment set in the back-door adjustment formula [Pearl, 1995, 2009], we have the following identification results.

**Lemma 3.5.** *With observational data, for $m \in \{1, \ldots, M\}$, if $\hat{Y} \notin pa(A, \mathcal{H}) \cup \mathbf{S}^{(m)}(A)$[3], then the post-intervention distribution can be calculated from the observational data by:*

$$P(\hat{Y} = y | do(A = a)) = \int P\left(\hat{Y} = y | A = a, pa(A, \mathcal{H}) \cup \mathbf{S}^{(m)}(A)\right) dP\left(pa(A, \mathcal{H}) \cup \mathbf{S}^{(m)}(A)\right)$$

$$= \int \frac{P\left(\hat{Y} = y, A = a | pa(A, \mathcal{H}) \cup \mathbf{S}^{(m)}(A)\right)}{P\left(A = a | pa(A, \mathcal{H}) \cup \mathbf{S}^{(m)}(A)\right)} dP\left(pa(A, \mathcal{H}) \cup \mathbf{S}^{(m)}(A)\right),$$

*where $P\left(A = a | pa(A, \mathcal{H}) \cup \mathbf{S}^{(m)}(A)\right)$ is estimated via $g(X|_{pa(A,\mathcal{H}) \cup \mathbf{S}^{(m)}(A)}; \hat{\phi}^{(m)})$ in Sec. 3.2.*

*Proof of Lemma 3.5.*

$$P(\hat{Y} = y | do(A = a))$$

$$= \int P\left(\hat{Y} = y | do(A = a), pa(A, \mathcal{H}) \cup \mathbf{S}^{(m)}(A)\right) dP\left(pa(A, \mathcal{H}) \cup \mathbf{S}^{(m)}(A)\right)$$

$$= \int P\left(\hat{Y} = y | A = a, pa(A, \mathcal{H}) \cup \mathbf{S}^{(m)}(A)\right) dP\left(pa(A, \mathcal{H}) \cup \mathbf{S}^{(m)}(A)\right)$$

$$= \int \frac{P\left(\hat{Y} = y, A = a | pa(A, \mathcal{H}) \cup \mathbf{S}^{(m)}(A)\right)}{P\left(A = a | pa(A, \mathcal{H}) \cup \mathbf{S}^{(m)}(A)\right)} dP\left(pa(A, \mathcal{H}) \cup \mathbf{S}^{(m)}(A)\right),$$

where the first and the third equations are from the conditional probability formula.

$\square$

From Lemma 3.5, for each possible parental set $pa(A, \mathcal{H}) \cup \mathbf{S}^{(m)}(A)$ with $m = 1, \ldots, M$, given estimated propensities $\hat{e}_i^{(m)}$ in Sec. 3.2, we can train a treatment effect estimation model $h(x_i; \hat{\psi}^{(m)}) = \hat{\tau}_i^{(m)}$ to estimate $P(\hat{Y} = y \mid do(A = 1)) - P(\hat{Y} = y \mid do(A = 0))$ by minimizing

$$\ell(\hat{\psi}^{(m)}) = \frac{1}{N} \sum_{i=1}^{N} \left( \frac{A_i f(x_i; \theta)}{\hat{e}_i^{(m)}} - \frac{(1 - A_i) f(x_i; \theta)}{1 - \hat{e}_i^{(m)}} - h(x_i; \hat{\psi}^{(m)}) \right)^2,$$

where $f(x_i; \theta) = \hat{Y}_i$ is an outcome predictor parameterized by $\theta$, and $h(x_i; \hat{\psi}^{(m)}) = \hat{\tau}_i^{(m)}$ aims to evaluate the interventional fairness violation of that learned predictor.

### 3.4 Min-Max Joint Learning Approach

We now aim to train a predictor to satisfy interventional fairness. Since the parental set of the sensitive attribute in the true DAG is unknown, we propose a min-max learning approach to control for the worst-case interventional fairness violations of the predictor. Specifically, given all possible causal effects $\hat{\tau}_i^{(m)}$ of the sensitive attribute $A$ on the predictor $\hat{Y}$ in Section 3.3, the prediction model $\hat{Y} = f(x; \theta)$ is trained by minimizing the average prediction error with the worst-case violations of interventional fairness as a penalty term

$$\min_{\theta} \ell(\theta; \hat{\psi}^{(1)}, \ldots, \hat{\psi}^{(M)}) = \frac{1}{N} \sum_{i=1}^{N} (Y_i - f(x_i; \theta))^2 + \gamma \cdot \max_{m} \frac{1}{N} \sum_{i=1}^{N} \xi_i^{(m)},$$

$$\text{s.t. } \hat{\tau}_i^{(m)} \leq C + \xi_i^{(m)}, \ i = 1, \ldots, N, \ m = 1, \ldots, M,$$

$$\hat{\tau}_i^{(m)} \geq -C - \xi_i^{(m)}, \ i = 1, \ldots, N, \ m = 1, \ldots, M,$$

$$\xi_i^{(m)} \geq 0, \ i = 1, \ldots, N, \ m = 1, \ldots, M,$$

---

[3]It is worth noting that this assumption always holds, since the training of the predictor $\hat{Y}$ cannot affect the origin sensitive attribute $A$.

---

**Algorithm 2:** A min-max optimization approach alternatively updating possible counterfactual treatment effect models and prediction model controlling the worse-case fairness violations.

---

**Input:** Sensitive attribute $A$, outcome of interest $Y$, and other observable attributes $X$, possible adjustment sets $\mathbf{S}^{(m)}$ and propensity model parameters $\hat{\phi}^{(m)}$ from Alg. 1.

---

1 **while** *stopping criteria is not satisfied* **do**
2    **for** $m = 1, \ldots, M$ **do**
3       **for** *number of steps for training the possible counterfactual treatment effect model* **do**
4          Sample a batch of units $\{(a_{m_k}, x_{m_k}, y_{m_k})\}_{k=1}^{K}$;
5          Update $\hat{\psi}^{(m)}$ by descending along the gradient $\nabla_{\hat{\psi}^{(m)}} \ell(\hat{\psi}^{(m)}; \theta)$;
6       **end**
7       Compute possible counterfactual treatment effects $\hat{\tau}_i^{(m)} = h(x_i; \hat{\psi}^{(m)})$;
8    **end**
9    **for** *number of steps for training the prediction model* **do**
10       Sample a batch of units $\{(a_l, x_l, y_l)\}_{l=1}^{L}$;
11       Update $\theta$ by descending along the gradient of min-max loss $\nabla_\theta \ell(\theta; \hat{\psi}^{(1)}, \ldots, \hat{\psi}^{(M)})$;
12    **end**
13 **end**

---

which is a convex optimization problem when $\hat{\tau}_i^{(m)} = h(x_i; \hat{\psi}^{(m)})$ is linear. It is equivalent to

$$\min_\theta \tilde{\ell}(\theta) = \frac{1}{N} \sum_{i=1}^{N} (Y_i - f(x_i; \theta))^2 + \lambda \cdot \max_m \frac{1}{N} \sum_{i=1}^{N} \left[ (-C - \hat{\tau}_i^{(m)})_+ + (\hat{\tau}_i^{(m)} - C)_+ \right],$$

where $\gamma$ and $\lambda$ are hyper-parameters for trade-off between prediction accuracy and interventional fairness. Since achieving strict interventional fairness for all individuals, i.e., having zero causal effects of sensitive attribute on the predictor, is usually unrealistic and would come at the cost of much prediction accuracy, we introduce a slack variable $\xi_i^{(m)}$ for each individual and a pre-specified threshold $C$, which penalizes the loss when the estimated causal effect $|\hat{\tau}_i^{(m)}| > C$. Note that when implementing the proposed min-max optimization approach, the treatment effect estimation models for evaluating the fairness violations in Section 3.3 and the prediction model controlling for worse-case fairness violations in Section 3.4 should be updated *alternatively*, which can be viewed as an iterative process of *interventional fairness evaluation* and *policy improvement* of the prediction model. We summarized the whole min-max optimization algorithm in Alg. 2.

## 4 Experiments

In this section, both synthetic and real-world experiments are conducted to evaluate the prediction accuracy and fairness of our approach. The root mean squared error (RMSE) between $Y$ and $\hat{Y}$ is used to measure the prediction performance, and the RMSE between $\hat{Y}|do(A = a)$ and $\hat{Y}|do(A = a')$ is used to measure the violation of the interventional fairness, named "unfairness".

**Baselines.** We consider six baseline prediction models: (1) **Full** uses all attributes, (2) **Unaware** uses all attributes except the sensitive attribute, (3) **Oracle** uses all attributes that are non-descendants of the sensitive attribute given the ground-truth DAG, (4) **FairRelax** uses all definite non-descendants and possible descendants of the sensitive attribute in a CPDAG (or an MPDAG), (5) **Fair** uses all definite non-descendants of the sensitive attribute in a CPDAG (or an MPDAG), and (6) $\epsilon$-**IFair** uses all attributes and implement the constrained optimization problem.

**Synthetic Study.** Synthetic data are generated from a linear structural equation model based on a ground-truth DAG. Specifically, we first randomly generate a DAG with $d$ nodes and $2d$ directed edges according to the Erdős-Rényi (ER) model with $d \in \{10, 20, 30, 40\}$ in our experiment. Following the previous studies [Zuo et al., 2022, 2024], the path coefficients $\beta_{jk}$ of directed edges $X_j \to X_k$ are sampled from a uniform distribution U($[-2, -0.5] \cup [0.5, 2]$). The data are generated using $X_k = \sum_{X_j \in pa(X_k)} \beta_{jk} X_j + \epsilon_i, i = 1, \ldots, n$, where $pa(X_k)$ represents the parent nodes of $X_k$, noise $\epsilon_i \sim N(0, \gamma)$ with $\gamma \in \{1.5, 2.5\}$, and $n$ is the sample size, which is set to 1,000 in our

Table 1: Average RMSE and unfairness for synthetic datasets on the held-out test set.

| **Noise = 1.5** | Node = 10, Edge = 20 | | Node = 20, Edge = 40 | | Node = 30, Edge = 60 | | Node = 40, Edge = 80 | |
|---|---|---|---|---|---|---|---|---|
| Method | RMSE ↓ | Unfairness ↓ | RMSE ↓ | Unfairness ↓ | RMSE ↓ | Unfairness ↓ | RMSE ↓ | Unfairness ↓ |
| Oracle | $0.757 \pm 0.349$ | $0.000 \pm 0.000$ | $0.579 \pm 0.245$ | $0.000 \pm 0.000$ | $0.571 \pm 0.194$ | $0.000 \pm 0.000$ | $0.578 \pm 0.200$ | $0.000 \pm 0.000$ |
| Full | $0.576 \pm 0.218$ | $0.195 \pm 0.232$ | $0.494 \pm 0.133$ | $0.095 \pm 0.128$ | $0.542 \pm 0.196$ | $0.063 \pm 0.083$ | $0.538 \pm 0.183$ | $0.067 \pm 0.113$ |
| Unaware | $0.587 \pm 0.219$ | $0.150 \pm 0.208$ | $0.498 \pm 0.134$ | $0.058 \pm 0.095$ | $0.544 \pm 0.196$ | $0.050 \pm 0.076$ | $0.540 \pm 0.183$ | $0.043 \pm 0.066$ |
| FairRelax | $0.653 \pm 0.256$ | $0.142 \pm 0.201$ | $0.586 \pm 0.217$ | $0.055 \pm 0.092$ | $0.603 \pm 0.241$ | $0.045 \pm 0.068$ | $0.611 \pm 0.254$ | $0.041 \pm 0.068$ |
| Fair | $0.747 \pm 0.293$ | $0.128 \pm 0.200$ | $0.627 \pm 0.223$ | $0.050 \pm 0.074$ | $0.661 \pm 0.263$ | $0.043 \pm 0.067$ | $0.630 \pm 0.292$ | $0.038 \pm 0.059$ |
| $\epsilon$-IFair | $0.644 \pm 0.262$ | $0.137 \pm 0.187$ | $0.570 \pm 0.215$ | $0.056 \pm 0.080$ | $0.589 \pm 0.239$ | $0.048 \pm 0.065$ | $0.609 \pm 0.241$ | $0.040 \pm 0.063$ |
| Ours | $0.623 \pm 0.210$ | $0.119 \pm 0.175$ | $0.561 \pm 0.126$ | $0.049 \pm 0.073$ | $0.597 \pm 0.185$ | $0.037 \pm 0.054$ | $0.606 \pm 0.178$ | $0.036 \pm 0.054$ |
| **Noise = 2.5** | Node = 10, Edge = 20 | | Node = 20, Edge = 40 | | Node = 30, Edge = 60 | | Node = 40, Edge = 80 | |
| Method | RMSE ↓ | Unfairness ↓ | RMSE ↓ | Unfairness ↓ | RMSE ↓ | Unfairness ↓ | RMSE ↓ | Unfairness ↓ |
| Oracle | $0.729 \pm 0.344$ | $0.000 \pm 0.000$ | $0.874 \pm 0.625$ | $0.000 \pm 0.000$ | $0.801 \pm 0.497$ | $0.000 \pm 0.000$ | $0.820 \pm 0.472$ | $0.000 \pm 0.000$ |
| Full | $0.667 \pm 0.274$ | $0.185 \pm 0.189$ | $0.761 \pm 0.440$ | $0.150 \pm 0.425$ | $0.736 \pm 0.417$ | $0.075 \pm 0.087$ | $0.729 \pm 0.334$ | $0.110 \pm 0.183$ |
| Unaware | $0.674 \pm 0.276$ | $0.065 \pm 0.094$ | $0.772 \pm 0.457$ | $0.062 \pm 0.126$ | $0.737 \pm 0.417$ | $0.032 \pm 0.043$ | $0.733 \pm 0.336$ | $0.041 \pm 0.079$ |
| FairRelax | $0.738 \pm 0.283$ | $0.059 \pm 0.077$ | $0.898 \pm 0.600$ | $0.050 \pm 0.119$ | $0.831 \pm 0.487$ | $0.030 \pm 0.040$ | $0.791 \pm 0.410$ | $0.040 \pm 0.079$ |
| Fair | $0.774 \pm 0.274$ | $0.052 \pm 0.067$ | $0.937 \pm 0.642$ | $0.046 \pm 0.118$ | $0.891 \pm 0.550$ | $0.029 \pm 0.039$ | $0.816 \pm 0.411$ | $0.039 \pm 0.079$ |
| $\epsilon$-IFair | $0.732 \pm 0.275$ | $0.055 \pm 0.082$ | $0.872 \pm 0.586$ | $0.046 \pm 0.101$ | $0.833 \pm 0.418$ | $0.025 \pm 0.045$ | $0.789 \pm 0.418$ | $0.039 \pm 0.078$ |
| Ours | $0.719 \pm 0.280$ | $0.049 \pm 0.073$ | $0.857 \pm 0.466$ | $0.045 \pm 0.090$ | $0.823 \pm 0.413$ | $0.023 \pm 0.031$ | $0.788 \pm 0.334$ | $0.038 \pm 0.070$ |

(a) RMSE     (b) Unfairness     (c) RMSE     (d) Unfairness

Figure 1: Performance under varying hyper-parameters $C$ and $\lambda$ on RMSE and unfairness.

experiment. Next, we use the PC algorithm in the causal-learn package to learn a CPDAG. Then we randomly select two nodes as the outcome $Y$ and the sensitive attribute $A$, respectively. We sample $A$ from a Bernoulli distribution with probability $\sigma(\sum_{X_j \in pa(A)} \beta_{jA} X_j + \epsilon_i)$, where $\sigma(\cdot)$ denotes the sigmoid function. The proportion of training data and test data are set to 0.8 and 0.2, respectively.

**Performance Comparison.** Table 1 shows the results of baselines and our approach. First, **Full** and **Unaware** perform better on RMSE, while **Fair**, **FairRelax**, $\epsilon$-**IFair**, and our approach have a significant advantage on unfairness. Note that our approach outperforms **Fair**, **FairRelax**, and $\epsilon$-**IFair** in all scenarios on both RMSE and unfairness metrics, because the proposed method makes predictions with all attributes and controls unfairness by the adjustment sets, whereas **Fair** and **FairRelax** can hardly find the true descendants of the sensitive attribute and $\epsilon$-**IFair** can hardly find the true causal effects when the learned CPDAG is not accurate in practice. In addition, Figure 1 shows the change in RMSE and unfairness as $C$ and $\lambda$ increase. When $C$ is increasing, RMSE is decreasing significantly, while unfairness is increasing. Because the larger $C$ is, the looser the control of causal effects, which is beneficial for prediction performance but hurts fairness. Similar arguments hold for $\lambda$, where a larger $\lambda$ will increase the cost of fairness violations in the optimization problem, thus benefiting fairness but hurting prediction accuracy.

**MPDAG with Background Knowledge.** Given the CPDAG, we randomly select a certain percentage of the directed edges in the true DAG as background knowledge and impose it on the already learned CPDAG. For example, if $A \rightarrow B$ is selected from the true DAG, we add this directed edge to the learned CPDAG regardless of the original relationship between $A$ and $B$ in the CPDAG to obtain an MPDAG and then adjust the MPDAG according to the Meek's rule. Figure 2 shows the effect of background knowledge on performance. As the background knowledge increases, the RMSE of **Fair** and **FairRelax** increases and the unfairness decreases significantly because more background knowledge forces **Fair** and **FairRelax** to have fewer nodes to make predictions. For our approach and $\epsilon$-**IFair**, both prediction and unfairness performance become better as the background knowledge ratio increases, which is attributed to the more accurate identification of the possible adjustment sets and the causal effect. Note that our approach stably outperforms $\epsilon$-**IFair** under varying background knowledge ratios. In addition, Table 2 reports the change of precision and recall for finding adjustment sets with increasing background knowledge ratio.

**Effect of Different Number of Parent Node.** We evaluate the RMSE and unfairness performance with varying numbers $p$ of parent nodes of the sensitive attribute. The results are shown in Table 3.

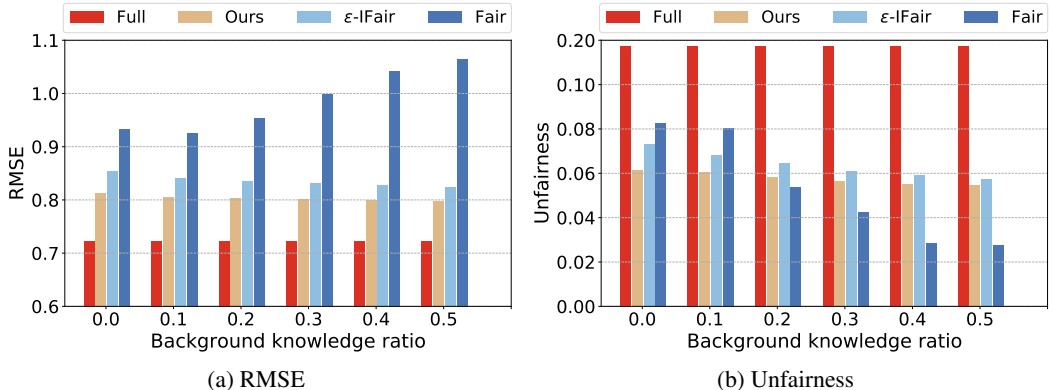

|  | (a) RMSE | (b) Unfairness |

Figure 2: RMSE and unfairness performance under varying background knowledge ratio.

Table 2: Average precision and recall for finding the adjustment sets in MPDAG.

| Background Knowledge | 0% | 10% | 20% | 30% | 40% | 50% |
|---|---|---|---|---|---|---|
| Precision ↑ | $0.438 \pm 0.489$ | $0.438 \pm 0.489$ | $0.466 \pm 0.492$ | $0.580 \pm 0.486$ | $0.642 \pm 0.471$ | $0.742 \pm 0.437$ |
| Recall ↑ | $0.265 \pm 0.353$ | $0.265 \pm 0.353$ | $0.274 \pm 0.350$ | $0.329 \pm 0.355$ | $0.353 \pm 0.347$ | $0.400 \pm 0.346$ |

Table 3: Average RMSE and unfairness for synthetic datasets on the held-out test set with different numbers $p$ of parent nodes of the sensitive attribute.

|  | Unaware | Fair | FairRelax | $\epsilon$-IFair | Ours |
|---|---|---|---|---|---|
| RMSE ($p = 0$) | $0.601 \pm 0.235$ | $0.642 \pm 0.241$ | $0.635 \pm 0.240$ | $0.633 \pm 0.237$ | $0.635 \pm 0.233$ |
| Unfairness ($p = 0$) | $0.063 \pm 0.082$ | $0.041 \pm 0.051$ | $0.045 \pm 0.050$ | $0.041 \pm 0.021$ | $0.035 \pm 0.016$ |
| RMSE ($p = 2$) | $0.528 \pm 0.298$ | $0.601 \pm 0.291$ | $0.597 \pm 0.294$ | $0.590 \pm 0.206$ | $0.584 \pm 0.201$ |
| Unfairness ($p = 2$) | $0.111 \pm 0.090$ | $0.099 \pm 0.075$ | $0.100 \pm 0.075$ | $0.100 \pm 0.088$ | $0.096 \pm 0.107$ |
| RMSE ($p = 4$) | $0.718 \pm 0.281$ | $0.860 \pm 0.261$ | $0.834 \pm 0.253$ | $0.818 \pm 0.237$ | $0.792 \pm 0.228$ |
| Unfairness ($p = 4$) | $0.129 \pm 0.073$ | $0.113 \pm 0.058$ | $0.120 \pm 0.050$ | $0.110 \pm 0.072$ | $0.103 \pm 0.086$ |

As $p$ increases, due to the presence of more backdoor paths, the RMSE performance and unfairness performance of all methods decreases, however, our method still outperforms the baseline methods.

**The Performance on the Classification Problem.** We conduct more experiments to examine if the proposed method can be applied to classification problems. Specifically, we modify the data generation process (DGP) to clip the outcome variable $Y$ to 1 if $Y > 0$, and to 0 if $Y \leq 0$, and the rest DGP remains the same. In this scenario, we adopt AUC as the evaluation metric instead of RMSE. The experiment results are shown in Table 4. We find that both **Full** and **Unaware** perform better on AUC, while **Fair**, **FairRelax**, $\epsilon$-**IFair**, and our approach perform better on unfairness. Note that our approach outperforms **Fair**, **FairRelax**, and $\epsilon$-**IFair** in all scenarios for both AUC and unfairness.

**Case Study.** The sensitive attributes contained in many widely used datasets for fair machine learning have no parent nodes, such as sex in the Adult dataset[4] [Kohavi, 1996] and race in the COMPAS dataset[5] [Angwin et al., 2022]. Because sex, race, and age cannot be affected by other collected features, we further consider the Open University Learning Analytics Dataset (OULAD) dataset[6] [Kuzilek et al., 2017], in which disability is treated as the sensitive attribute and final_grade is treated as the outcome. The COMPAS dataset contains 6,172 units with 9 attributes such as gender and number_of_priors, the Adult dataset contains 48,842 units with 14 attributes such as age, education, and race, and the OULAD dataset contains 32,593 units with 11 attributes including demographic information such as gender, age, education_level, etc. For this case study, we first learn a CPDAG from the raw data using the PC algorithm in the causal-learn package and obtain an MPDAG with the background knowledge such as sex can not be caused by other attributes. Second,

---

[4]https://archive.ics.uci.edu/dataset/2/adult

[5]https://www.kaggle.com/datasets/danofer/compass

[6]https://www.archive.ics.uci.edu/dataset/349/open+university+learning+analytics+dataset

Table 4: Average AUC and unfairness for synthetic datasets on the test set on classification problem.

| Noise = 2.5 | NODE = 10, EDGE = 20 | | NODE = 20, EDGE = 40 | | NODE = 30, EDGE = 60 | | NODE = 40, EDGE = 80 | |
|---|---|---|---|---|---|---|---|---|
| Method | AUC ↑ | Unfairness ↓ | AUC ↑ | Unfairness ↓ | AUC ↑ | Unfairness ↓ | AUC ↑ | Unfairness ↓ |
| Oracle | $0.815 \pm 0.094$ | $0.000 \pm 0.000$ | $0.805 \pm 0.148$ | $0.000 \pm 0.000$ | $0.819 \pm 0.149$ | $0.000 \pm 0.000$ | $0.818 \pm 0.087$ | $0.000 \pm 0.000$ |
| Full | $0.845 \pm 0.071$ | $0.038 \pm 0.051$ | $0.889 \pm 0.083$ | $0.126 \pm 0.115$ | $0.855 \pm 0.111$ | $0.090 \pm 0.087$ | $0.842 \pm 0.086$ | $0.143 \pm 0.106$ |
| Unaware | $0.843 \pm 0.070$ | $0.017 \pm 0.021$ | $0.886 \pm 0.080$ | $0.105 \pm 0.152$ | $0.853 \pm 0.114$ | $0.076 \pm 0.082$ | $0.837 \pm 0.090$ | $0.113 \pm 0.121$ |
| FairRelax | $0.825 \pm 0.057$ | $0.017 \pm 0.021$ | $0.857 \pm 0.130$ | $0.084 \pm 0.148$ | $0.845 \pm 0.117$ | $0.074 \pm 0.106$ | $0.824 \pm 0.082$ | $0.112 \pm 0.119$ |
| Fair | $0.819 \pm 0.060$ | $0.015 \pm 0.021$ | $0.779 \pm 0.200$ | $0.082 \pm 0.143$ | $0.844 \pm 0.116$ | $0.074 \pm 0.106$ | $0.822 \pm 0.128$ | $0.094 \pm 0.122$ |
| $\epsilon$-IFair | $0.843 \pm 0.049$ | $0.018 \pm 0.017$ | $0.883 \pm 0.081$ | $0.081 \pm 0.087$ | $0.843 \pm 0.115$ | $0.068 \pm 0.088$ | $0.833 \pm 0.085$ | $0.098 \pm 0.105$ |
| Ours | $0.844 \pm 0.051$ | $0.015 \pm 0.016$ | $0.886 \pm 0.077$ | $0.080 \pm 0.130$ | $0.855 \pm 0.109$ | $0.069 \pm 0.094$ | $0.835 \pm 0.088$ | $0.089 \pm 0.119$ |

Table 5: Real-world experiment results on the COMPAS, Adult, and OULAD datasets. For the COMPAS dataset, the sensitive attribute is race, and for the Adult dataset, the sensitive attribute is sex. Both of the sensitive attributes have no parent nodes. For the OULAD dataset, the sensitive attribute is disability, which can have parent nodes.

| COMPAS | Full | Unaware | FairRelax | Fair | $\epsilon$-IFair | Ours |
|---|---|---|---|---|---|---|
| RMSE | $0.256 \pm 0.022$ | $0.261 \pm 0.023$ | $0.261 \pm 0.023$ | $0.263 \pm 0.022$ | $0.235 \pm 0.091$ | $0.219 \pm 0.020$ |
| Unfairness | $0.273 \pm 0.048$ | $0.269 \pm 0.052$ | $0.260 \pm 0.045$ | $0.238 \pm 0.045$ | $0.190 \pm 0.120$ | $0.179 \pm 0.011$ |

| Adult | Full | Unaware | FairRelax | Fair | $\epsilon$-IFair | Ours |
|---|---|---|---|---|---|---|
| RMSE | $0.433 \pm 0.024$ | $0.436 \pm 0.024$ | $0.607 \pm 0.131$ | $0.611 \pm 0.128$ | $0.413 \pm 0.010$ | $0.375 \pm 0.019$ |
| Unfairness | $0.506 \pm 0.021$ | $0.409 \pm 0.029$ | $0.209 \pm 0.205$ | $0.187 \pm 0.205$ | $0.155 \pm 0.009$ | $0.171 \pm 0.021$ |

| OULAD | Full | Unaware | FairRelax | Fair | $\epsilon$-IFair | Ours |
|---|---|---|---|---|---|---|
| RMSE | $0.502 \pm 0.041$ | $0.502 \pm 0.042$ | $0.503 \pm 0.041$ | $0.503 \pm 0.041$ | $0.499 \pm 0.041$ | $0.491 \pm 0.040$ |
| Unfairness | $0.088 \pm 0.024$ | $0.031 \pm 0.058$ | $0.029 \pm 0.023$ | $0.029 \pm 0.023$ | $0.027 \pm 0.020$ | $0.024 \pm 0.018$ |

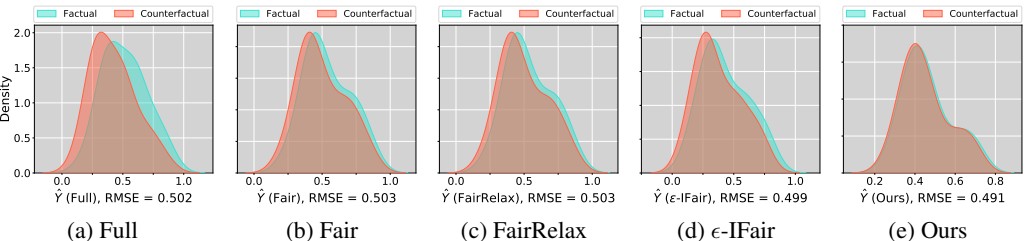

(a) Full    (b) Fair    (c) FairRelax    (d) $\epsilon$-IFair    (e) Ours

Figure 3: Density plot of the predicted $\hat{Y}|do(A = a)$ and $\hat{Y}|do(A = a')$ on OULAD data.

we randomly generate a DAG as the ground-truth from the learned MPDAG. After obtaining the DAG, we then divide the data into 100 random batches, and for each batch, we learn an MPDAG from the observed data and background knowledge. The path coefficients are determined based on linear regression and regard the residual of the regression as noise. The subsequent steps are the same as in the synthetic study. The experiment results are shown in Table 5, with density plots in Figure 3. First, both $\epsilon$-**IFair** and our method demonstrate the superiority of our approach in both prediction performance and fairness compared to other baselines. In addition, our method stably outperforms $\epsilon$-**IFair**, further validating the effectiveness of the proposed min-max joint learning approach.

## 5 Conclusion

This paper aims to achieve interventional fairness from observational data when the causal graph is unknown or partially known. Interestingly, we show it is actually sufficient to enumerate all possible parental sets of the sensitive attributes via a local approach, instead of enumerating all DAGs at high computational cost. We then propose a general min-max optimization framework to achieve interventional fairness that is easy applicable to CPDAGs and maximally oriented PDAGs (MPDAGs) with the added background knowledge. One limitation of our approach is due to the proposed approach relying on a CPDAG given by the causal discovery algorithm and estimations of the propensities, which may lead to mild violations of interventional fairness when the CPDAG or estimates are inaccurate. Another possible limitation, which also serves as a future research direction, is to achieve interventional fairness in the presence of hidden variables with partially known DAGs.

## Acknowledgments and Disclosure of Funding

The authors thank the anonymous reviewers for their valuable comments. This work was supported in part by National Natural Science Foundation of China (623B2002, 12201629). Yue Liu was supported by the MOE Project of Key Research Institute of Humanities and Social Sciences (22JJD110001).

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

## Broader Impacts

This paper proposes a general min-max optimization framework that can effectively achieve interventional fairness when the true causal graph is unknown or partially known. In contrast to statistical fairness, interventional fairness considers possible counterfactual decision-makings, not just based on the observed data. The implications of achieving interventional fairness in algorithms where the true causal graph is unknown or partially known are mainly in the following aspects. First, reducing bias: machine learning models may learn and reflect bias in the training data and unconsciously apply this bias to individuals in their predictions. interventional fairness helps reduce this possible bias. Second, improve fairness: if we can achieve interventional fairness on partially known causal graphs, our models will be prevented from treating unfairly because of the sensitive attributes. Third, enhancing trust: as our algorithms are able to process data in a fairer way, people's trust in those algorithms increases. This is critical in many areas, such as healthcare, finance, and justice. Fourth, promote policy making: understanding and addressing interventional fairness issues in algorithms can help policy makers better understand and regulate these technologies to ensure their fairness and transparency in practice. In a nutshell, studying how to effectively achieve interventional fairness in scenarios such as unknown causal graphs or the presence of hidden variables is both challenging and socially significant, and deserves more effort.

## A  More Discussion on the Previous Work

To tackle the above problem, a recent work [Zuo et al., 2022] proposes to use observed data to first classify variables into three categories: *definite non-descendants*, *possible descendants*, and *definite descendants* of the sensitive attributes. Next, by noting that a prediction model would be counterfactually fair if the prediction model is a function of the non-descendants of sensitive attributes [Kusner et al., 2017], as shown in Table 6, FAIR method is proposed to use only the definite non-descendants, and FAIRRELAX method further incorporates possible descendants.

Table 6: Comparison of methods to achieve interventional fairness from PDAGs. Both FAIR and FAIRRELAX employ a two-stage approach: they first learn a CPDAG from observed data, and then make prediction with the definite non-descendants (and possible descendants) of the sensitive attribute. Our method alternatively updates the predictions using *all* variables and possible counterfactual treatment effects via a min-max optimization.

| Variables (example in Figure 4(b)) | FAIR | FAIRRELAX | OURS |
|---|---|---|---|
| Definite non-descendants ($\emptyset$) | ✓ | ✓ | ✓ |
| Possible descendants ($X_1, X_2$) | ✗ | ✓ | ✓ |
| Definite descendants ($\emptyset$) | ✗ | ✗ | ✓ |

However, both FAIR and FAIRRELAX forbid the use of definite descendants for model prediction, which greatly compromises the prediction accuracy. In particular, the sensitive attribute is usually an inherent nature of data, making most of the attributes are its descendants [Wu et al., 2019a].

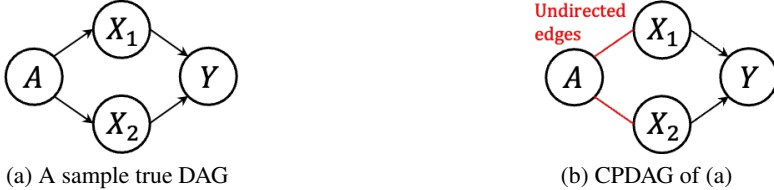

(a) A sample true DAG              (b) CPDAG of (a)

Figure 4: A toy example for illustration: FAIR has *no available variables* for prediction; FAIRRELAX uses $\{X_1, X_2\}$ *without* further fairness constraint; OURS uses $\{A, X_1, X_2\}$ *with* a min-max constraint bounding all possible counterfactual treatment effect.

We proceed with a toy example for illustration: Figure 4(a) shows a sampled DAG as the ground-truth, and given the observed data, FAIR and FAIRRELAX algorithms first learn a Markov equivalence

**Algorithm 3:** A local algorithm for finding possible adjustment sets and estimating corresponding propensity model parameters of the sensitive attribute $A$.

**Input:** Sensitive attribute $A$, CPDAG $\mathcal{G}^*$.

1  Set $\mathcal{S}_A = \emptyset$ and $m = 1$;

2  **for** *each* $\mathbf{S}^{(m)} \subset \mathrm{sib}(A, \mathcal{G}^*)$ *such that orienting* $\mathbf{S}^{(m)} \to A$ *and* $A \to \mathrm{sib}(A, \mathcal{G}^*) \backslash \mathbf{S}^{(m)}$ *does not introduce any v-structure collided on* $A$ **do**

3      **for** *number of steps for training the possible propensity model on* $\mathbf{S}^{(m)}$ **do**

4          Sample a batch of units $\{(a_{m_k}, x_{m_k}|_{\mathbf{S}^{(m)}})\}_{k=1}^{K}$;

5          Update $\hat{\phi}^{(m)}$ by descending along the gradient $\nabla_{\hat{\phi}^{(m)}} \ell(\hat{\phi}^{(m)}; \mathbf{S}^{(m)})$;

6      **end**

7      $\mathcal{S}_A \leftarrow \mathcal{S}_A \cup \mathbf{S}^{(m)}$ and $m \leftarrow m + 1$;

8  **end**

**Output:** A set $\mathcal{S}_A$ of possible adjustment sets $\mathbf{S}^{(m)}$ and propensity model parameters $\hat{\phi}^{(m)}$.

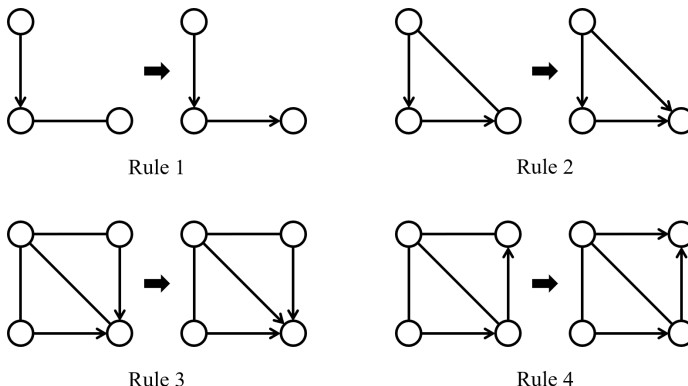

Figure 5: The visualization of four rules of Meek's criteria. If the graph on the left-hand side of a rule is an induced subgraph of a PDAG $\mathcal{G}$, then orient the undirected edge such that the resulting subgraph is the one on the right-hand side of the rule.

class of DAGs that encode the same set of conditional independencies from the data, also known as a completely partially directed acyclic graph (CPDAG), as shown in Figure 4(b). One on hand, the *definite non-descendants* of sensitive attribute $A$ is a empty set, thus FAIR is unable to return valid prediction model. On the other hand, the *possible descendants* of the sensitive attribute $A$ are $\{X_1, X_2\}$, thus FAIRRELAX uses both $X_1$ and $X_2$ to predict $Y$ by minimizing the empirical risk *without imposing any further fairness constraint*. However, such relaxation would lead to a serious violation of interventional fairness, due to the nodes used ($X_1$ and $X_2$ in this example) for outcome regression might be descendants of the sensitive attribute $A$ in the true DAG, as in Figure 4(a).

When the true causal graph is unknown, to the best of our knowledge, Zuo et al. [2022] performed the first work to obtain an interventional fairness predictor on an MPDAG, which focuses on utilizing the properties of the causal graph (Level 1 in Kusner et al. [2017]) – to make predictions with the definite non-descendants (and possible descendants) of the sensitive attribute, as shown in Table 6. However, further incorporating the descendants of sensitive attribute into the predictor may also achieve interventional fairness by "cancelling out" the counterfactual treatment effects, which utilizes the observed variables more sufficiently and further improves the accuracy of the prediction.

We provide an intuition for the rationality and the advantages of using all variables by adopting the toy example in Figure 4. Suppose the structural equations in Figure 4(a) are: $A = U_A$, $X_1 = A + U_1$, $X_2 = A + U_2$, and $Y = 2X_1 + X_2 + U_Y$, which satisfies the faithfulness assumption [Uhler et al., 2013]. In such a case, as discussed before, the FAIR algorithm proposed in Zuo et al. [2022] prevents all variables from predicting $Y$, while the FAIRRELAX algorithm uses both $X_1$ and $X_2$ to predict $Y$ without imposing any fairness constraints, and therefore cannot achieve causal fairness, since $X_1$ and $X_2$ are descendants of $A$. To achieve more accurate predictions with interventional

**Algorithm 4:** Constructing the MPDAG $\mathcal{H}$ from CPDAG $\mathcal{G}^*$ and $\mathcal{B}_d$ Using Meek's Rules

**Input:** A CPDAG $\mathcal{G}^*$, a set of directed edges $\mathcal{B}_d$.
**Output:** An MPDAG $\mathcal{H}$ or FAIL.

1 Set $\mathcal{H} = \mathcal{G}^*$;
2 **while** $\mathcal{B}_d \neq \emptyset$ **do**
3     Choose an edge $u \to v$ from $\mathcal{B}_d$;
4     $\mathcal{B}_d = \mathcal{B}_d \setminus \{u \to v\}$;
5     **if** $u \to v$ *or* $u - v$ *is in* $\mathcal{H}$ **then**
6        Orient $u \to v$ in $\mathcal{H}$;
7        Orient the other edges under the Meek's rules in Figure 5;
8     **else**
9        **return** FAIL;
10     **end**
11 **end**
12 **return** MPDAG $\mathcal{H}$;

---

fairness guarantees, one may notice that a function of $X_1 - X_2$ can be used to predict $Y$. On the one hand, this is strictly counterfactually fair due to the fact that $X_1 - X_2 = U_1 - U_2$, which is independent of the sensitive attribute $A$. On the other hand, this is informative for predicting $Y$ due to $\mathrm{Cov}(X_1 - X_2, Y) = 2\,\mathrm{Var}(U_1) - \mathrm{Var}(U_2) \neq 0$.

However, the true DAG and the corresponding structural equations are unknown in many real-world scenarios, which poses a great challenge to estimate the possible counterfactual treatment effects. To address this problem, an intuitive approach is to first find a Markov equivalence class over all vertices, which can be achieved using standard causal discovery methods, e.g., PC [Spirtes et al., 2000] and GES [Chickering, 2002b], and then to globally enumerate all the possible DAGs in the equivalence class and estimate their causal effects for each. However, as discussed in Section 7 of Zuo et al. [2022], this intuitive way to enumerate all DAGs is computationally expensive and unrealistic.

