# OpenReview forum: "A Local Method for Satisfying Interventional Fairness with Partially Known Causal Graphs"
_NeurIPS.cc/2024/Conference — NeurIPS 2024 poster_

### Official Review · Reviewer_GJNz · 2024-06-28

**Soundness:** 3
**Presentation:** 3
**Contribution:** 3
**Rating:** 5
**Confidence:** 4

**Summary:**

This paper develops a method for satisfying interventional fairness when the causal graph is partially known. The paper first develops a method for checking possible parental sets as well as a method for estimating propensities given a possible parental set. Then, the interventional distribution is computed using the parental set as a back-door adjustment set. Finally, a min-max joint learning approach is proposed to learn a fair model based on the worst-case violation of interventional fairness.

**Strengths:**

Originality
The paper extends the previous methods to MPDAGs and develops a novel method for learning fair models based on the worst-case violation of interventional fairness.

Quality
The proposed method is theoretically sound. It builds upon established theoretical results from previous works while also providing rigorous theoretical analysis of the newly proposed approaches.

Clarity
The main results are summarized as lemmas. However, a lemma is usually used to help prove a larger theorem. The authors may change lemmas to propositions.

Significance
The experiment results show that the proposed method can achieve good fairness and accuracy performance without providing exact causal graphs.

**Weaknesses:**

This paper presents a solid work. The main reason I am inclined towards rejection is the applicability of this work. The proposed method relies on the existence of parent nodes of the sensitive attribute. However, as also mentioned in the paper, in most datasets the sensitive attributes have no parent nodes. To be more specific, most sensitive attributes have no parents in nature. For example, common sensitive attributes like sex/gender, race, nationality, etc., are determined at birth. As a result, the proposed method is applicable to very limited sensitive attributes like disability, which significantly limits the importance of this work. I recommend that the authors may apply the proposed method to more general causal inference scenarios where the treatment node may generally have parent nodes.

**Questions:**

In the experiments, the accuracy is measured by RMSE. Can the proposed method be applied to classification problems?

===============post-rebuttal comments============

The authors partially addressed my concerns, and I value the contributions made by the paper. I have raised my score.

---

> ### Author Rebuttal · Authors · 2024-08-07
>
> We sincerely appreciate the reviewer’s great efforts and insightful comments to improve our manuscript. We are also encouraged for the reviewer to recognize the **Originality**, **Quality**, **Clarity**, and **Significance** of our work. In below, we would like to address the reviewer's concerns regarding the **Applicability** of our proposed method from both theoretical and experimental aspects.
>
> ***
>
> ### **Main Result 1: The proposed method can also be applied in scenarios where sensitive attributes are identified as root nodes (W1).**
>
> - We **agree** with the reviewer that most sensitive attributes have no parents in nature. However, we respectively **disagree** that the proposed method relies on the existence of parent nodes of the sensitive attribute.
>
> ### **Theoretical Analysis**
>
> - For our studied interventional fairness, if the sensitive attribute $A$ is as a root node in the causal DAG, then there will be **no backdoor path** from the sensitive attribute $A$ to the outcome variable $Y$.
>
> - At this point, **the causal estimands in the definition of interventional fairness and our method degenerate to the correlation estimands.**
>
> - Compared to previous Fair, FairRelax, and $\epsilon$-IFair methods, **our method doesn't rely on variable selection and strong causal assumptions**, thus still should be considered as a more plausible method.
>
> ### **More Real-World Experiments**
>
> - We **add experiments on two widely used real-world datasets**: the **Adult** and the **COMPAS** datasets, where **the sensitive attributes are "sex" and "race" as the root nodes**, respectively.
>
> |COMPAS|Full|Unaware|FairRelax|Fair|ε-IFair|Ours|
> | :--: | :--: | :--: | :--: | :--: | :--: | :--: |
> |RMSE|0.256 ± 0.022|0.261 ± 0.023|0.261 ± 0.023|0.263 ± 0.022|0.235 ± 0.091|0.219 ± 0.020 |
> |Unfairness|0.273 ± 0.048|0.269 ± 0.052|0.260 ± 0.045|0.238 ± 0.045|0.190 ± 0.120|0.179 ± 0.011|
> ||
>
> |Adult|Full|Unaware|FairRelax|Fair|ε-IFair|Ours|
> | :--: | :--: | :--: | :--: | :--: | :--: | :--: |
> |RMSE|0.433 ± 0.024|0.436 ± 0.024|0.607 ± 0.131|0.611 ± 0.128|0.413 ± 0.010|0.375 ± 0.019|
> |Unfairness|0.506 ± 0.021|0.409 ± 0.029|0.209 ± 0.205|0.187 ± 0.205|0.155 ± 0.009|0.171 ± 0.021|
> ||
>
> - From the above experimental results, we find that **our method still outperforms previous methods when the sensitive attribute is used as the root node**.
>
>
> ### **More Synthetic Experiments with Varying Number of Parent Nodes**
>
> - **To further explore the impact of the number of parent nodes of sensitive attributes on the accuracy and fairness performance, we also added more synthetic experiments with varying number of parent nodes.**
>
> |Metrics|Unaware|Fair|FairRelax|$\epsilon$-IFair|Ours|
> |:--:|:--:|:--:|:--:|:--:|:--:|
> |RMSE (p = 0)|0.601 ± 0.235|0.642 ± 0.241|0.635 ± 0.240|0.633 ± 0.237|0.635 ± 0.233|
> |Unfairness (p = 0)|0.063 ± 0.082|0.041 ± 0.051|0.045 ± 0.050|0.041 ± 0.021|0.035 ± 0.016|
> |RMSE (p = 2)|0.528 ± 0.298|0.601 ± 0.291|0.597 ± 0.294|0.590 ± 0.206|0.584 ± 0.201|
> |Unfairness (p = 2)|0.111 ± 0.090|0.099 ± 0.075|0.100 ± 0.075|0.100 ± 0.088|0.096 ± 0.107|
> |RMSE (p = 4)|0.718 ± 0.281|0.860 ± 0.261|0.834 ± 0.253|0.818 ± 0.237|0.792 ± 0.228|
> |Unfairness (p = 4)|0.129 ± 0.073|0.113 ± 0.058|0.120 ± 0.050|0.110 ± 0.072|0.103 ± 0.086|
> ||
>
> - **As the number of parent nodes increases, the advantage of our method over **Fair**, **FairRelax**, and **$\epsilon$-IFair** gradually increases.** This is because when there are more parent nodes, there are more backdoor paths, thus our method can find these backdoor paths to better control the causal effect of the sensitive attribute on the outcome variable.
>
> ***
>
> ### **Main Result 2: The proposed method can also be applied to the Classification Problems (Q1).**
>
> ### **Real-World Experiments**
>
> - For real-world experiments, we would like to kindly remind the reviewer that **the outcome variable is binary in the OULAD dataset** adopted in our original manuscript. **The outcome variables in our added experiments on the Adult and COMPAS datasets are also binary.**
>
> ### **Synthetic Experiments**
>
> - For synthetic experiments, as suggested by the reviewer, **we conduct more experiments using AUC as the evaluation metric instead of RMSE**. Specifically, we modify the data generation process (DGP) to clip the outcome variable $Y$ to 1 if $Y > 0$, and to 0 if $Y < 0$, and the rest DGP remains the same. The experiment results are shown below.
>
> |Noise = 2.5|Node = 10, Edge = 20 (AUC) |Node = 10, Edge = 20 (Unfairness) |Node = 40, Edge = 80 (AUC)|Node = 40, Edge = 80 (Unfairness)|
> |:--:|:--:|:--:|:--:|:--:|
> |Oracle|0.815 ± 0.094|0.000 ± 0.000|0.828 ± 0.087|0.000 ± 0.000|
> |Full|0.845 ± 0.071|0.038 ± 0.051|0.842 ± 0.086|0.143 ± 0.106|
> |Unaware|0.843 ± 0.070|0.017 ± 0.021|0.837 ± 0.090|0.113 ± 0.121|
> |FairRelax|0.825 ± 0.057|0.017 ± 0.021|0.824 ± 0.082|0.112 ± 0.119|
> |Fair|0.819 ± 0.060|0.015 ± 0.021|0.822 ± 0.128|0.094 ± 0.122|
> |$\epsilon$-IFair|0.843 ± 0.049|0.018 ± 0.017|0.833 ± 0.085|0.098 ± 0.105|
> |Ours|0.844 ± 0.051|0.015±0.016|0.835 ± 0.088|0.089 ± 0.119|
> ||
>
> - From the above results, **Full** and **Unaware** perform better on AUC, while **Fair**, **FairRelax**, **$\epsilon$-IFair**, and our approach perform better on unfairness. Note that our approach outperforms **Fair**, **FairRelax**, and **$\epsilon$-IFair** in all scenarios on both AUC and unfairness.
>
> ***
>
> ### **(Minor) Clarity Issues (S3)**
>
> > Clarity: The main results are summarized as lemmas. However, a lemma is usually used to help prove a larger theorem. The authors may change lemmas to propositions.
>
> - We thank the reviewer for raising such useful suggestions. In our revised version, we have changed all *Lemma*s to *Proposition*s.
>
> ***
>
> **We hope the above discussion will fully address your concerns about the applicability of our work, and we would really appreciate it if you could consider to raise your score.** We look forward to your insightful and constructive responses to further help us improve the quality of our work. Thank you!

---

> > ### Author Response · Authors · 2024-08-13
> > **We would like to summarize our efforts and changes during rebuttal as follows.**
> >
> > Dear Reviewer GJNz,
> >
> > Once again, we are grateful for your time and effort for reviewing our paper. Since the discussion period will end in around a day, we are very eager to get your feedback on our response. We understand that you are very busy, but we would highly appreciate it if you could take into account our response when updating the rating and having a discussion with AC and other reviewers.
> >
> > We are encouraged by your kind words supporting the **Originality, Quality, Clarity, and Significance** of our work. Meanwhile, it seems that your only concern currently is around the **Applicability** of our method. To facilitate checking, we are happy to summarize our efforts during rebuttal as follows.
> >
> > - **We theoretically showed that our proposed method can also works well in scenarios where sensitive attributes are as root nodes.**
> >
> > - **We added experiments on two widely used real-world datasets: the Adult and the COMPAS datasets, where the sensitive attributes are "sex" and "race" as the root nodes, respectively.**
> >
> > - **We claimed that our proposed method can also be applied to the classification problems, and highlighted that the current real-world experiment is conducted on the OULAD dataset, where the outcome variable is binary.**
> >
> > - **As suggested by the reviewer, we conduct more synthetic experiments using AUC as the evaluation metric instead of RMSE.**
> >
> > - **Benefiting from the reviewer, we have changed all _Lemmas_ to _Propositions_ to improve the presentation clarity.**
> >
> > - (Reviewer BRmE) **We added more simulation experiments to explore the effect of the inaccuracy of CPDAG on the performance of our method, as shown in Fig. 1 in the Supplementary PDF.**
> >
> > ***
> >
> > As the discussion deadline approaches, we are wondering whether our responses have properly addressed your concerns? Your feedback would be extremely helpful to us. If you have further comments or questions, we hope for the opportunity to respond to them.
> >
> > Many thanks,
> >
> > Submission18019 Authors

---

> > > ### Comment · Reviewer_GJNz · 2024-08-13
> > >
> > > I apologize for the late response.
> > >
> > > I agree with the authors' statement: if the sensitive attribute $A$ is a root node in the causal DAG, then there will be no backdoor path from the sensitive attribute $A$ to the outcome variable $Y$. In this case, the interventional fairness will degenerate to the correlation estimands.
> > >
> > > Could the authors elaborate a bit more on the advantages of the proposed method in the degenerated case mentioned above? As we know, common correlation-based fairness notions include demographic parity, risk ratio, equal opportunity, etc., and much research has been conducted to achieve these fairness notions. I wonder how the proposed method compares to these existing approaches in terms of factors like effectiveness, applicability, etc.

---

> ### Author Response · Authors · 2024-08-14
>
> First, we clarify that **methods for achieving fairness notions can be divided into the following three categories.**
>
> |Methods| Input(s) | Fairness notions | Limitation |
> |:--:|:--:|:--:|:--:|
> |achieve correlation-based fairness | observational data | demographic parity, risk ratio, equal opportunity, etc. | **cannot achieve causal fairness** |
> |achieve causal fairness **with** a known causal DAG | observational data, **a known causal DAG** |interventional fairness, counterfactual fairness, path-specific counterfactual fairness, etc. | **need strong prior knowledge for a known causal DAG** |
> |achieve causal fairness **without** a known causal DAG (ours) | observational data | interventional fairness, counterfactual fairness, path-specific counterfactual fairness, etc. | **with observational data, we can only know the Markov equivalence class, not the unique DAG** |
> ||
>
> **1. After careful thinking, we respectfully believe that it is not fair to compare our approach with methods for achieving correlation-based fairness in our degenerated case (in which the causal is the same as correlation), due to the following reasons.**
>
>   - For methods achieving correlation-based fairness, **their applicability usually fails to achieve causal fairness.** However, **our approach can achieve causal fairness with general applicability,** where it degenerates to correlation-based fairness only in a specific case (in which the sensitive attribute is a root node in the causal DAG).
>   - The most important point, from the table above, is **how do we determine if the sensitive attribute is the root node in the true causal DAG when we only have the observational data?**
> - We kindly invite the reviewer to refer to a toy example in Figure 4 on page 13 of our manuscript. Specifically, Figure 4(a) is the true causal DAG, in which the sensitive attribute is a root node. However, as shown in Figure 4(b), **with only observation data, we cannot distinguish from the following causal relations: $X_1 \leftarrow A \to X_2$, or $X_1 \to A \to X_2$, or $X_1 \leftarrow A \leftarrow X_2$,** because they are all in the same Markov equivalence class! But **only the first causal relation** $X_1 \leftarrow A \to X_2$ corresponds to the case where **the sensitive attribute is a root node!**
>
> **2. Meanwhile, we agree with the reviewer that it is fair to compare our approach with methods for achieving causal fairness _without_ a known causal DAG, in terms of factors like effectiveness, applicability, etc., especially at the degenerated case.**
>
> **To the best of our knowledge, Fair [1], FairRelax [1], and $\epsilon$-IFair [2] are all the existing prior works focusing on the same problem** (causal fairness with partially known causal graph). To clarify the **core idea, key assumptions, limitations, sensitivity to wrong CPDAG, and how they perform in the degenerated case in which the sensitive attribute as the root node**, we summarize our findings as follows.
>
> |Methods| Core idea | Key assumptions | Limitations | Sensitivity to wrong CPDAG | Sensitive attribute as the root node |
> |:--:|:--:|:--:|:--:|:--:|:--:|
> |Fair [1] and FairRelax [1] | two-phase approach: first obtains a CPDAG from causal discovery methods, then uses definite descendants (and possible descendants) of the sensitive attribute to train a fair model | a correct CPDAG, and **very strong assumption that the sensitive attribute is a root node for identification** | only uses partial variables, instead of all variables, may **significantly reduce the prediction accuracy** | **very high**, the effectiveness of such approaches completely rely on the correctness of the causal discovery algorithm to find the correct CPDAG | **poor fairness performance**, because the second phase does not impose any causal fairness constraint |
> |$\epsilon$-IFair [2] | a **global approach** by enumerating **all subclasses of DAGs where the causal effect is identifiable** | a correct CPDAG, and **strong assumptions that there is no undirect edge between sensitive features and non-sensitive features** | **high time complexity** | **weaker** than that of Fair [1] and FairRelax [1] (see Figure 1 in supplementary PDF) | causal effect is always identifiable in such a case, and **enumerating all DAGs globally needs much time** |
> | Ours | a **local approac**h by enumerating **all possible parental sets of the sensitive attribute** | a correct CPDAG **(with no other strong assumptions, because we intend to do partially identification, not fully identification as in [1] and [2])** | **need a extra propensity model** for fair training | **weaker** than that of Fair [1] and FairRelax [1] (see Figure 1 in supplementary PDF) | causal effect is always identifiable in such a case, and **enumerating locally needs less time** |
> ||

---

> ### Author Response · Authors · 2024-08-14
>
> - **In the degenerated case, we conclude that Fair [1] and FairRelax [1] perform poorly in both accuracy and fairness.**
>   - On one hand, they only use partial variables, not all, which reduces their accuracy performance.
>   - On the other hand, after selecting the definite descendants (and possible descendants) of the sensitive attribute, they does not impose any causal fairness constraint, which reduces their fairness performance.
>   - If the sensitive attribute is a root node in the causal DAG, most of the variables would be identified as the definite descendants (and possible descendants) via the causal discovery algorithms, which increases the availability of more variables and makes the unfair problem be more serious.
>   - We also validate such claims on our added experiments on the Adult and the COMPAS datasets during rebuttal.
> - **In the degenerated case, both of the $\epsilon$-IFair [2] and our approaches can work well, but the time complexity for implementing $\epsilon$-IFair [2] in much larger than that of our approach.**
>   - This is because $\epsilon$-IFair [2] enumerates all subclasses of DAGs where the causal effect is identifiable, and when the sensitive attribute is the root node, the causal effect is naturally identifiable, so $\epsilon$-IFair [2] enumerates almost all possible DAGs.
>   - For our local method, we only enumerate all possible parental sets of the sensitive attribute, whose enumerating space is much smaller than that of $\epsilon$-IFair [2], especially when we have plenty nodes in the real-world datasets.
>
> ***
>
> **We sincerely hope that the reviewer can carefully go over our discussion above -- we really spent a lot of time trying to make it clearer!** We are also very grateful for the multiple rounds of communication with you, as this level of engagement has significantly contributed to improving the quality and positioning of our work (in the extensive fairness literature).
>
> As the discussion deadline approaches, please let us know if you have any further feedback. We truly appreciate your beneficial comments as well as your consideration to upgrade your score -- thank you so much!!
>
> ***
>
> **References**
>
> [1] Counterfactual fairness with partially known causal graph. NeurIPS, 2022.
>
> [2] Interventional fairness on partially known causal graphs: A constrained optimization approach. ICLR, 2024.

---

> > ### Author Response · Authors · 2024-08-14
> >
> > Dear Reviewer GJNz,
> >
> > Since the discussion period will end in a few hours, we will be online waiting for your feedback on our rebuttal, which we believe has fully addressed your concerns.
> >
> > We understand that you are very busy, but we would highly appreciate it if you could take into account our response when updating the rating and having a discussion with AC and other reviewers.
> >
> > Thanks for your time,
> >
> > Submission18019 Authors

---

### Official Review · Reviewer_BRmE · 2024-07-08

**Soundness:** 4
**Presentation:** 4
**Contribution:** 3
**Rating:** 8
**Confidence:** 3

**Summary:**

In this paper, the author(s) talk about a better way to estimate interventional fairness in the case where we have partially known Directed Acyclic Graphs (DAG). It lists out how most previous works have only done this for fully known DAG's and recent attempts to address the case of unknown graphs have utilized strong assumptions and or reduced prediction accuracy.
The authors introduce  a min-max optimization framework to achieve interventional fairness and boost accuracy and test it on both generated data and real-world data.

**Strengths:**

The paper does a good assessment of former work on the topic and builds directly on it.
The terms and definitions and clearly outlined and explained adequately for easy understanding. The authors put their theory into practice and using both synthetic and real-world data.

**Weaknesses:**

The paper's approach relies heavily on the accuracy of the CPDAG and on the generated propensity scores. The paper could address measures to address interventional fairness in the case of inaccuracies coming from the CPDAG

**Questions:**

CPDAG was used on line 57 but was but the full meaning was not given until line 96.

**Limitations:**

The author brings it up clearly in their conclusion that their method relies on Completely Partially Directed Acyclic Graph (CPDAG) and on the propensity scores returned and therefore any errors in these can affect their method.
Also, the case where we have latent variables or confounders was not addressed.

---

> ### Author Rebuttal · Authors · 2024-08-07
>
> We sincerely appreciate the reviewer’s great efforts and insightful comments to improve our manuscript. We are also encouraged by the very positive comments on our paper.
>
> > The paper's approach relies heavily on the accuracy of the CPDAG and on the generated propensity scores. The paper could address measures to address interventional fairness in the case of inaccuracies coming from the CPDAG.
>
> As suggested by the reviewer, we added more simulation experiments to explore the effect of the degree of inaccuracy of CPDAG on the performance of our method. The experimental results are shown in Fig. 1 in the Supplementary PDF.
>
> > CPDAG was used on line 57 but was but the full meaning was not given until line 96.
>
> We thank the reviewer for pointing out the typo. In our revised manuscript, we will provide the full name of CPDAG and more explanation at Line 57.
>
> > Flag For Ethics Review: Ethics review needed: Data privacy, copyright, and consent
>
> We're guessing you misclicked the flag for ethics review. We would like to clarify that all used datasets in our original manuscript and rebuttal are public and widely adopted datasets.
>
> - The OULAD dataset: https://www.archive.ics.uci.edu/dataset/349/open+university+learning+analytics+dataset
>
> - The Adult dataset: https://archive.ics.uci.edu/dataset/2/adult
>
> - The COMPAS dataset: https://www.kaggle.com/datasets/danofer/compass
>
> ***
>
> We hope the above discussion will fully address your concerns about our work. We look forward to your insightful and constructive responses to further help us improve the quality of our work. Thank you!

---

> > ### Comment · Reviewer_BRmE · 2024-08-12
> >
> > I have fully read your rebuttal and I agree to them and yes I misclicked the flag for ethics review

---

> > > ### Author Response · Authors · 2024-08-12
> > > **Thank you for agreeing with our rebuttal and for the clarification on misclicking the ethical review.**
> > >
> > > We are thankful that the reviewers took the time to fully read our rebuttal and found our rebuttal meaningful. We appreciate your very positive evaluation of our work. Benefiting from your comments, the added experiments about the performance under inaccurate CPDAG can further enhance the quality of our manuscript -- thank you so much!!

---

### Official Review · Reviewer_3Hwm · 2024-07-12

**Soundness:** 4
**Presentation:** 3
**Contribution:** 4
**Rating:** 8
**Confidence:** 4

**Summary:**

This work investigates interventional fairness given partially known causal graphs. Compared to existing methods, it employs all variables and does not need to rely on additional strong assumptions for identification. Specifically, it offers a min-max optimization framework which produces counterfactual fairness in maximally oriented PDAGs. Experiments on synthetic and real-world datasets showcase the efficacy of the proposed method.

**Strengths:**

The manuscript is commendably clear in its presentation, particularly in articulating the research gap. This clarity facilitates a quick comprehension of the central theme and the contributions of the study.

The toy examples provided are meticulously crafted, serving effectively to elucidate several underlying concepts integral to the research.

The application of a min-max optimization algorithm is a highlight of this work, elegantly addressing the research problems identified.

**Weaknesses:**

Overall, the paper is well-composed, substantiated by robust experimental outcomes and comprehensive theoretical analysis. However, some sections could benefit from further clarification to enhance understandability. Specific points of ambiguity are addressed below:

•	The statement “One may notice that a function of X1 − X2 can be used to predict Y” appears somewhat abrupt and disrupts the flow of reading. A detailed explanation or a graphical illustration might clarify the independence between U1 − U2 and A.

•	At line 165, the rationale behind transforming certain undirected edges into directed edges in the opposite direction remains unclear. Elaboration on this would be beneficial.

•	The term ‘direct causal information set’ warrants a clear, explicit definition within the main text to avoid potential misunderstandings.


---

Thank you very much for providing thorough answers to my queries, which have satisfactorily addressed my concerns. I am pleased to adjust my score.

**Questions:**

See the weakness above.

**Limitations:**

The limitation has been well discussed in the conclusion part.

---

> ### Author Rebuttal · Authors · 2024-08-07
>
> We sincerely appreciate the reviewer’s great efforts and insightful comments to improve our manuscript. In below, we address these concerns point by point and try our best to update the manuscript accordingly.
>
> > **The statement “One may notice that a function of X1 − X2 can be used to predict Y” appears somewhat abrupt and disrupts the flow of reading. A detailed explanation or a graphical illustration might clarify the independence between U1 − U2 and A.**
>
> **Response:** We thank the reviewer for the helpful comments. We will add the following toy example to illustrate such claims.
>
> - Let $A = U_A, X_1 = A + U_1, X_2 = A + U_2$, and $Y = 2X_1 + X_2 + U_Y$.
> - Because $Cov(X_1 – X_2, Y) \neq 0$, thus $X_1 – X_2$ is useful in predicting $Y$.
> - Meanwhile, $X_1 – X_2 = U_1 – U_2$ is a constant for any value of $A$, so $U_1 – U_2$ is independent of $A$.
> - Thus, using $X_1 – X_2$ as the variable for the prediction can still achieve counterfactual fairness.
>
> > **At line 165, the rationale behind transforming certain undirected edges into directed edges in the opposite direction remains unclear. Elaboration on this would be beneficial.**
>
> **Response:** We would like to clarify the need for ‘transforming certain undirected edges into directed edges’ as follows.
>
> - We know that causal discovery algorithms such as the PC algorithm can only return CPDAGs with undirected edges, not DAGs where all edges are directed.
>
> - By definition, $sib(A)$ is a set containing all nodes that have an undirected edge with $A$, and our algorithm needs to find the set of all possible parent sets of the sensitive attribute $A$.
>
> - Thus, we need to enumerate all possible cases (i.e., for some $Z \in sib(A), Z \rightarrow A$, and for other $Z \in sib(A), A \rightarrow Z$), and determine whether a new v-structure is generated.
>
> - Formally, let $\mathbf{S}(A)$  be a subset of $sib(A)$, we can obtain a DAG from CPDAG by changing all undirected edges $\\{Z-A, \forall Z\in \mathbf{S}(A)\\}$ into the directed edges $\\{Z\to A, \forall Z\in \mathbf{S}(A)\\}$ as parents, and all of other undirected edges $\\{Z- A, \forall Z \not\in \mathbf{S}(A)\\}$ into the directed edges with opposite direction $\\{Z\leftarrow A, \forall Z \not\in \mathbf{S}(A)\\}$ as children.
>
> > **The term ‘direct causal information set’ warrants a clear, explicit definition within the main text to avoid potential misunderstandings.**
>
> **Response:** The ‘direct causal information set’ is assumed to be the direct causal information in the form $X \to Y$, indicating that $X$ is a direct cause of $Y$. Such direct causal information are usually determined by the prior knowledge. Meanwhile, it is worth noting that other forms of background knowledge, such as tier orderings, specific model restrictions, or data obtained from previous experiments, can also induce MPDAGs [1-4]. As suggested by the reviewer, we will add the above explicit definition within the main text to avoid potential misunderstandings.
>
> ***
>
> **References**
>
> [1] Alain Hauser and Peter Buhlmann. Characterization and greedy learning of interventional Markov equivalence classes of directed acyclic graphs. JMLR, 2012.
>
> [2] Marco F Eigenmann, Preetam Nandy, and Marloes H Maathuis. Structure learning of linear gaussian structural equation models with weak edges. ArXiv:1707.07560, 2017.
>
> [3] Yuhao Wang, Liam Solus, Karren Yang, and Caroline Uhler. Permutation-based causal inference algorithms with interventions. NeurIPS, 2017.
>
> [4] Dominik Rothenhausler, Jan Ernest, and Peter Buhlmann. Causal inference in partially linear structural equation models. The Annals of Statistics, 2018.
>
> ***
>
> **We hope the above discussion will fully address your concerns about our work.** We look forward to your insightful and constructive responses to further help us improve the quality of our work. Thank you!

---

> > ### Author Response · Authors · 2024-08-12
> > **Thanks for raising your score to "Strong Accept"!**
> >
> > We thank the reviewer for helping us to improve the presentation clarity of our manuscript -- we will definitely incorporate all the clarifications into our final version. We are happy that you are willing to improve your score from 7 to 8. Thanks!!

---

### Official Review · Reviewer_t9F5 · 2024-07-13

**Soundness:** 3
**Presentation:** 3
**Contribution:** 3
**Rating:** 6
**Confidence:** 2

**Summary:**

This paper aims for interventional fairness with sufficient prediction accuracy by employing a min-max optimization framework. The proposed approach aims for partially directed acyclic graphs (PDAGs) and extends itself to maximally oriented PDAGs (MPDAGs). Finally, the approach is evaluated on synthetic and real-world datasets.

**Strengths:**

The authors addressed a nice and interesting fairness problem in this paper. The proposed approach seemed to be theoretically robust. They also provided detailed empirical analysis.

**Weaknesses:**

Below I share some weaknesses.

* Above line 188: It is unclear what $S^{(i)}(A); 1<=i<=M$ refers to. The authors should explain the superscripts.
* Equations should be numbered. The Equation after line 192 should be explained in more detail.
* Line 201: The authors should provide some intuition/explanation about the backdoor adjustment formula.
* It seems that there are no baselines based on prior works. Is there no work that solves the fairness problem with causal effect estimation?

**Questions:**

Below I provide my questions:
* Line 178: why can it not be $X_i \leftarrow A \rightarrow X_j$?
* What is the computational complexity of the proposed approach?
* How can the proposed approach be connected with the min-max approach proposed by Xia et al [1] and the invariant prediction problem proposed by Subbaswamy et al [2]


[1] Xia, Kevin, et al. "The causal-neural connection: Expressiveness, learnability, and inference." Advances in Neural Information Processing Systems 34 (2021): 10823-10836.\
[2] Subbaswamy, Adarsh, Peter Schulam, and Suchi Saria. "Preventing failures due to dataset shift: Learning predictive models that transport." The 22nd International Conference on Artificial Intelligence and Statistics. PMLR, 2019.

**Limitations:**

The authors properly discussed their limitations.

---

> ### Author Rebuttal · Authors · 2024-08-07
>
> We sincerely appreciate the reviewer’s great efforts and insightful comments to improve our manuscript. In below, we address these concerns point by point and try our best to update the manuscript accordingly.
>
> ## Weaknesses
>
> > **Above line 188 : It is unclear what $S^{(i)}(A) ; 1<=i<=M$ refers to. The authors should explain the superscripts.**
>
> - We know that causal discovery algorithms can only return CPDAGs with undirected edges, not DAGs where all edges are directed.
>
> - Let $sib(A)$ be a set containing all nodes that have an undirected edge with $A$, and our algorithm needs to find the set of all possible parent sets of the sensitive attribute $A$.
>
> - $S^{(i)}(A) ; 1<=i<=M$ is the subsets of $sib(A)$, such that $\mathcal{S}_A=\\{p a(A)\cup \mathbf{S}^{(1)}(A), \ldots, p a(A)\cup \mathbf{S}^{(M)}(A)\\}$ consists of all possible parent sets of the sensitive attribute $A$ obtained from our algorithm.
>
> > **Equations should be numbered. The Equation after line 192 should be explained in more detail.**
>
> - The propensity $P(A \mid p a(A)\cup \mathbf{S}^{(m)}(A))$ is defined as the probability of the (binary) sensitive attribute $A$ given its (possible) parent set $pa(A)\cup \mathbf{S}^{(m)}(A)$.
>
> - The Equation after line 192 is the binary classification cross-entropy loss for training the propensity model $g(X; \hat \phi^{(m)})$ for estimating the true propensity $P(A \mid p a(A)\cup \mathbf{S}^{(m)}(A))$.
>
> - The input is covariates $X$ restricted on the possible parent set $p a(A, \mathcal{H})\cup \mathbf{S}^{(m)}(A)$, denoted as $X|_{p a(A, \mathcal{H})\cup \mathbf{S}^{(m)}(A)}$.
>
> > **Line 201: The authors should provide some intuition/explanation about the backdoor adjustment formula.**
>
> - Whenever we undertake to evaluate the effect of one factor $A$ on another $Y$, the question arises as to whether we should adjust our measurements for possible variations in some other factors $X$, otherwise known as "confounders".
>
> - The backdoor adjustment formula aims to partitioning the population into groups that are homogeneous relative to $X$, assessing the effect of $A$ on $Y$ in each homogeneous group, and then averaging the results.
>
> - Formally, the backdoor adjustment formula is $P(Y=y \mid do(A=a))=\sum_x P(Y=y \mid A=a, X=x) P(X=x)$, which computes the association between $A$ and $Y$ for each value $x$ of $X$, then averages over those values, also referred as "adjusting for $X$".
>
> > **It seems that there are no baselines based on prior works. Is there no work that solves the fairness problem with causal effect estimation?**
>
> - For compared baselines, we would like to clarify that FairRelax [R1], Fair [R1], and $\epsilon$-IFair [R2] are all prior works solving the same problem (causal fairness with partially known causal graph). We will explicitly add citations in our revised "Baselines" part.
>
> - To the best of our knowledge, we have compared the most comprehensive and SOTA methods, e.g., [R2] is published in ICLR 24.
>
> - The important reason for the limited baselines is because most previous causal fairness work requires known causal graphs (DAGs), whereas we only need tabular data (from which we can only obtain CPDAGs).
>
> ## Questions
>
> > **Line 178: why can it not be $X_i \leftarrow A \to X_j$?**
>
> - From Definition 3.1 in Line 157, the v-structure is defined as $X_i \to A \leftarrow X_j$, not $X_i \leftarrow A \to X_j$.
>
> - From Lemma 3.2 in Line 170, a set $\mathbf{S}(A) \subset sib(A)$ is a possible parent set of the sensitive attribute $A$, if and only if there is no more v-structures.
>
> - Thus, as stated in Line 178, if $X_i$ and $X_j$ are not adjacent, they cannot be both in the parent set of $A$. Otherwise, a new v-structure $X_i \to A \leftarrow X_j$ is formed.
>
> > **What is the computational complexity of the proposed approach?**
>
> - First, our proposed approach mainly includes the following 3 steps: constructing the CPDAG from the tabular data, estimating all possible propensities (Alg. 1), and learning fair classifier via minimax approach (Alg. 2).
>
> - From [R1], the complexity of constructing the CPDAG in the worst case is $\mathcal{O}(|sib(S, \mathcal{G})+ch(S, \mathcal{G})| *|E(\mathcal{G})|)$, where $|E(\mathcal{G})|$ is the number of edges in $\mathcal{G}$.
>
> - For Alg. 1 and Alg. 2, the proposed local method needs to identify the v-structures for every node pairs in $sib(S, \mathcal{G})$, thus the computational complexity is $\mathcal{O}(|sib(S, \mathcal{G})|^2)$.
>
> - Thus, the overall computational complexity is $\mathcal{O}(|sib(S, \mathcal{G})+ch(S, \mathcal{G})| *|E(\mathcal{G})|+|sib(S, \mathcal{G})|^2)$, which also illustrates the advantage of our local method instead of enumerating on all possible DAGs.
>
> > **How can the proposed approach be connected with the min-max approach proposed by Xia et al [1] and the invariant prediction problem proposed by Subbaswamy et al [2]?**
>
> - For the min-max approach proposed by Xia et al [1], one similar point is that the input is also the observational data, rather than requiring the known DAG.
>
> - They found that a neural net is unable to predict the effects of interventions given observational data alone (Thm. 1), which motivates to introduce **a special type of SCM called a neural causal model (NCM)** for performing causal inferences.
>
> - However, our paper still aims to conduct the causal inferences **in the context of general SCM**, with minimax approach for bounding the uncertainties of the true DAGs.
>
> - For the invariant prediction problem proposed by Subbaswamy et al [2], they aims to solve the **dataset shift problem with different training and target distributions.**
>
> - However, our study aims to achieve causal fairness without the explicit knowledge of causal DAGs, with **same training and target distributions.**
>
>
> ****
>
> **References**
>
> [R1] Counterfactual fairness with partially known causal graph. NeurIPS, 2022.
>
> [R2] Interventional fairness on partially known causal graphs: A constrained optimization approach. ICLR, 2024.

---

> > ### Comment · Reviewer_t9F5 · 2024-08-12
> >
> > Thank you for the rebuttal. The authors have addressed my concerns. Also, after reading other reviewers' responses, I have decides to increase my score.

---

> > > ### Author Response · Authors · 2024-08-13
> > > **We are happy that your concerns have been addressed. Thanks for raising your score!**
> > >
> > > We're glad our rebuttal addressed your concerns. Thank you for your helpful comments -- it helps make our manuscripts easier to follow. May I know if our clarifications can improve your ```Confidence: 2``` in your review? We will definitely put our discussions into our final version -- thank you so much!!

---

### Author Rebuttal · Authors · 2024-08-07

Dear all reviewers and AC,

Please kindly find the attachment as our added one page experimental results.

Thanks,

Authors from Submission #18019

---

### Decision · Program_Chairs · 2024-09-25

**Decision:**

Accept (poster)

**Comment:**

The authors proposed a method for achieving interventional fairness when the causal graph is unknown.
The proposed method enumerates only those equivalent PDAGs that affect interventional fairness among the equivalent causal DAGs, and then trains a model to minimize the worst-case fairness within these equivalent PDAGs.
Through discussions with the reviewers, the authors clarified the effectiveness of the proposed method and its broad applicability.
Although there were comments from the reviewers regarding the differences in assumptions between the proposed method and existing methods, as well as their performance comparison, the authors responded to these points appropriately. Incorporating the content of these discussions into the manuscript would further improve the quality of the paper.